# LEARNING A GAME BY PAYING THE AGENTS

**Brian Hu Zhang**[*]
Massachusetts Institute of Technology
zhangbh@csail.mit.edu

**Tao Lin**[*]
Microsoft Research &
The Chinese University of Hong Kong, Shenzhen
lintao@cuhk.edu.cn

**Yiling Chen**
Harvard University
yiling@seas.harvard.edu

**Tuomas Sandholm**
Carnegie Mellon University &
Strategy Robot, Inc.,
Strategic Machine, Inc.,
Optimized Markets, Inc
sandholm@cs.cmu.edu

## ABSTRACT

We study the problem of learning the utility functions of no-regret learning agents in a repeated normal-form game. Differing from most prior literature, we introduce a principal with the power to observe the agents playing the game, send agents signals, and give agents *payments* as a function of their actions. We show that the principal can, using a number of rounds polynomial in the size of the game, learn the utility functions of all agents to any desired precision $\varepsilon > 0$, for *any* no-regret learning algorithms of the agents. Our main technique is to formulate a zero-sum game between the principal and the agents, where the principal chooses strategies among the set of all payment functions to minimize the agent's payoff. Finally, we discuss implications for the problem of *steering* agents. We introduce, using our utility-learning algorithm as a subroutine, the first algorithm for steering arbitrary no-regret learning agents to a desired equilibrium without prior knowledge of their utility functions.

## 1 INTRODUCTION

Most literature on game theory aims to understand the behavior of agents in a game given the preferences of the agents. That is, *knowing* what the agents want, how will they act? In this paper, we consider the inverse of this problem: if all we can observe is how agents act, can we infer what they want, that is, their utility functions? The problem of infering agents' utility functions from their behavior is known as "learning from revealed preferences" (e.g., Beigman & Vohra (2006); Zadimoghaddam & Roth (2012)) or "inverse game theory" (e.g., Kuleshov & Schrijvers (2015)). Despite a long history, these two lines of work have some limitations. First, they often assume that the observed behavior of the agents is *(Nash) equilibrium behavior*. This is arguably unrealistic, especially because Nash equilibria are PPAD-hard to compute (Daskalakis et al., 2006; Chen et al., 2009). Second, they mostly focus on *passive* problems, aiming to learn agents' utility functions from a fixed set of behavioral data. Often, this creates trivial impossibility results, stemming from the fact that the behavioral data available is simply not enough to determine the agents' preferences.

In this paper, we consider an *active, non-equilibrium* inverse game problem. A principal observes the actions of agents playing an unknown repeated game, and seeks to learn the agents' utility functions from those observations. We do not assume the agents to play equilibria. Instead, they can use *any no-regret algorithms* to learn to choose actions. The principal can actively modify the unknown game by giving *payments* to the agents as a function of their play, as well as providing *signals*, in the spirit of correlated equilibria (Aumann, 1974). The signaling scheme and payment scheme can change from round to round, depending on the agents' past actions. Under this setup, we ask:

*Can a principal learn the utility functions of no-regret learning agents?*

---

[*]Equal contribution. Order determined randomly.

We will give a positive answer to this question, by designing an algorithm for the principal to learn the game via payments and signals. Then, we will apply our algorithm to *steering* no-regret learning agents toward desirable outcomes, a problem introduced by Zhang et al. (2024). Building on their results, we will show that it is possible to steer agents to optimal outcomes even without prior knowledge of their utility functions.

## 1.1 OVERVIEW OF OUR RESULTS

In our model, agents play a fixed normal-form game $\Gamma$ repeatedly over $T$ rounds, using arbitrary no-regret learning algorithms. A principal, initially knowing nothing about the agents' utility functions, can give non-negative *payments* to the agents, that get added to the agents' utilities, to influence the agents' behavior. The principal aims to learn the utility functions of all agents to a given target precision $\varepsilon > 0$, based on the actions taken by the agents.

Learning the utility functions completely is impossible, because agents' incentives only depend on *relative* utilities between their actions, not *absolute* utilities. Thus, we only demand that the principal output utility functions that yield a *strategically-equivalent* game, that is, one in which all agents' relative utilities are identical to those in the true game. Equivalently, we identify each agent's utility function up to a term that does not depend on that agent's action.

Our main result is an efficient algorithm for the principal to learn the game by paying the agents. Let $n$ be the number of agents, $m_i$ be the number of actions of each agent $i$, and $M = \prod_i m_i$ be the total number of action profiles. Assume that each agent's regret in $T$ rounds is at most $\mathcal{O}(\sqrt{T})$ (satisfied by typical no-regret algorithms). Our algorithm learns the game in polynomially many rounds:

**Theorem 1.1** (Informal version of Theorems 4.3 and 4.4)**.** *There exists a principal algorithm that learns a game to precision $\varepsilon$ in $M^{\mathcal{O}(1)}/\varepsilon^2$ rounds. This is tight up to the exponent hidden by the $\mathcal{O}$.*

The main idea of our algorithm is surprisingly simple but powerful. In the single-agent case, we let the principal play a zero-sum game with the agent, where the agent chooses actions to maximize its reward, which is utility plus payment, while the principal chooses payments to minimize the agent's reward. This game admits a unique equilibrium where the principal's payment function is equal to the negation of the agent's utility function. By running no-regret algorithms against each other, the principal and agent can reach such an equilibrium, so the negative average payment function will be an accurate estimate of the agent's utility function. In the multi-agent case, we use signals to separate the learning problems for different agents, reducing the problem to the single-agent case.

We then turn to a motivating application, which is the problem of *steering* no-regret learners to desirable outcomes, introduced by Zhang et al. (2024). Departing from them, we do not require the principal to have prior knowledge of the agents' utility functions. We define a solution concept called *correlated equilibrium with payments* (CEP), in which the principal has a utility function, and wishes to optimize its utility minus the amount of payment that it must give. Departing from Zhang et al. (2024) again, it is possible for the total amount of payment to be nonzero in equilibrium (*i.e.*, linearly increasing in $T$), so long as the corresponding increase in principal utility is large enough to justify the payments. We then show that the principal-optimal CEP exactly characterizes the value (averaged across timesteps) that the principal can achieve in the limit $T \to \infty$:

**Theorem 1.2** (Informal version of Theorems 5.2 and 5.3)**.** *Let $F^*(\Gamma)$ be the objective value for the principal in the principal-optimal CEP in game $\Gamma$. Then, against no-regret agents,*

- *even knowing the game $\Gamma$ exactly, the principal cannot achieve time-averaged value better than $F^*(\Gamma) + \mathsf{poly}(M) \cdot T^{-1/2}$, and*

- *with no prior knowledge of the agents' utilities, there exists an algorithm for the principal to achieve time-averaged value at least $F^*(\Gamma) - \mathsf{poly}(M) \cdot T^{-1/4}$.*

All our algorithms are implementable by the principal in $\mathsf{poly}(M)$ running time. To our knowledge, our Result 1 is the first positive result for utility-learning with arbitrary no-regret agents. Our Result 2 is the first in steering any no-regret agents without prior knowledge of their utilities, and the first exact characterization of the optimal value achievable by the principal in the steering problem.

## 1.2 RELATED RESEARCH

*Steering agents to achieve desirable outcomes* is an important subject of study in economics, computer science, and control theory (e.g., Monderer & Tennenholtz (2003); Zhang et al. (2024); Canyakmaz et al. (2024); Yao et al. (2025)). In particular, Zhang et al. (2024) introduced the problem of steering no-regret learners via payments. Critically, most prior works on steering assume that the principal knows the agents' utility functions. We study how the agents' utility functions can be learned, the solution to which will serve for any downstream applications including steering.

Besides the aforementioned works, another literature about learning agents' utility functions from their behavior is *learning in Stackelberg games*, where the principal cannot pay the agents but can influence the agents' actions by changing its own strategy, and the agents respond myopically (Letchford et al., 2009; Peng et al., 2019) or by learning algorithms (Haghtalab et al., 2022). Strong impossibility results are known for this problem: without regularity conditions or without knowing the details of the agents' learning algorithms, the principal cannot learn the agent's utility function (Zhang et al., 2025). In contrast, we consider a setting where the principal gives payments but does not take actions. We show that the use of payment makes a significant difference: the principal can now learn the utility functions of any no-regret agents without knowing their behavioral details.

Even with payment, the problem of learning from learning agents is still challenging. As we will discuss more in Section 4, a key obstacle is the non-forgetfulness of agents' no-regret algorithms. The payment given to the agents in the past affect their future behavior. Non-forgetfulness is a known issue in multi-agent learning dynamics (Wu et al., 2022; Cai et al., 2024; Scheid et al., 2024). We overcome this obstacle by designing a zero-sum-game-based learning algorithm for the principal and using signals to influence agents, without requiring the agents' algorithms to be forgetful.

Our use of signals to influence agents is inspired by *information design* (*e.g.*, Kamenica & Gentzkow (2011)). In fact, Feng et al. (2022); Bacchiocchi et al. (2024); Alanqary et al. (2026) studied how to learn agents' utility functions by providing signals to shape agents' beliefs and behaviors in repeated games. While they consider myopically best-responding agents, we allow any no-regret learning agents, a more challenging setting necessitating the combination of signals and payments.

The literature on *playing against no-regret agents* studies how the principal should play a game if they know the agents' utilities and some properties of the agents' algorithms (*e.g.*, Braverman et al. (2018); Deng et al. (2019); Mansour et al. (2022); D'Andrea (2023); Lin & Chen (2025); Arunachaleswaran et al. (2025)). For example, Deng et al. (2019) show that, if agents run *mean-based* no-regret algorithms, then the principal can gain *more* than the Stackelberg value in a Stackelberg game. Our algorithms and setting, on the other hand, consider *worst-case* no-regret agent behaviors. While most of the cited papers consider principal-agent problems with a single agent and with no payment, we consider arbitrary payment-argumented multi-agent games.

## 2 PRELIMINARIES

**Notations.** Throughout this paper, $\tilde{\mathcal{O}}$ and $\tilde{\Omega}$ hide factors logarithmic in their argument. The symbol $[n]$ denotes the set of positive integers $\{1, \ldots, n\}$. The notation $f \lesssim g$ means $f \leq \mathcal{O}(g)$, and $f \gtrsim g$ means $f \geq \Omega(g)$. For a vector $\boldsymbol{v} \in \mathbb{R}^m$, its $i$-th component is denoted by $\boldsymbol{v}[i]$ or $v_i$. Vector of ones and zeros are $\boldsymbol{1} = (1, \ldots, 1)$ and $\boldsymbol{0} = (0, \ldots, 0)$. $\mathbb{I}\{\cdot\}$ is the indicator function, *i.e.*, for a statement $p$, $\mathbb{I}\{p\} = 1$ if $p$ is true and $0$ if $p$ is false.

**Normal-form games.** A normal-form game $\Gamma = (n, A, U)$ consists of a set of $n$ *agents*, or *players*, which we will identify with the set of integers $[n]$. Each agent $i$ has an action set $A_i$ of size $m_i \geq 2$. We will let $m := \max_i m_i$ and $M = \prod_i m_i$. A tuple $\boldsymbol{a} \in A := A_1 \times \cdots \times A_n$ is an *action profile*. Each agent has a *utility function* $U_i : A \to [0, 1]$, denoting the utility for agent $i$ when the agents play action profile $\boldsymbol{a} \in A$. A *mixed strategy* of agent $i$ is a distribution $\boldsymbol{x}_i \in \Delta(A_i)$. We will overload the utility function $U_i$ to accept mixed strategies, so that $U_i(\boldsymbol{x}_1, \ldots, \boldsymbol{x}_n)$ is the expected utility for agent $i$ when every agent $j \in [n]$ independently samples $a_j \sim \boldsymbol{x}_j$. As is standard in game theory, we use $-i$ to refer to the tuple of all agents except $i$. For instance, $U_i(a_i', \boldsymbol{a}_{-i})$ is the utility of agent $i$ when agent $i$ plays action $a_i' \in A_i$ and other agents play $\boldsymbol{a}_{-i} \in A_{-i} = \bigtimes_{j \neq i} A_j$.

**No-regret learning.** In *no-regret learning*, a learner has a convex compact strategy set $\mathcal{X} \subset \mathbb{R}^m$, and interacts with a possibly adversarial environment. On each timestep $t$, the learner selects a strategy $\boldsymbol{x}^t \in \mathcal{X}$. Simultaneously, the environment, possibly adversarially, selects a linear utility vector $\boldsymbol{u}^t \in \mathbb{R}^m$, which we assume to be bounded: $|\langle \boldsymbol{u}^t, \boldsymbol{x} \rangle| \lesssim 1$. The learner's *regret* after $T$ timesteps is $R(T) = \max_{\boldsymbol{x} \in \mathcal{X}} \sum_{t=1}^{T} \langle \boldsymbol{u}^t, \boldsymbol{x} - \boldsymbol{x}^t \rangle$. We say that the learner is running a *no-regret algorithm* if its average regret $R(T)/T \to 0$ in the limit $T \to \infty$ for all possible sequences $(\boldsymbol{u}^1, \ldots, \boldsymbol{u}^T)$.

A well-known no-regret algorithm for general strategy set $\mathcal{X}$ is *projected gradient descent/ascent* (Zinkevich, 2003). Projected gradient ascent selects $\boldsymbol{x}^1 \in \mathcal{X}$ arbitrarily, and for every timestep $t > 1$ sets $\boldsymbol{x}^t = \Pi_{\mathcal{X}}[\boldsymbol{x}^{t-1} + \eta \boldsymbol{u}^{t-1}]$ where $\Pi_{\mathcal{X}}$ is the $\ell_2$-projection operator into $\mathcal{X}$. With step size $\eta = B/(G\sqrt{T})$, where $B$ is the $\ell_2$-diameter of $\mathcal{X}$ and $G$ bounds the $\ell_2$-norm of $\boldsymbol{u}^t$, the algorithm achieves a regret of $R(T) \lesssim BG\sqrt{T}$.

To apply no-regret learning to games, each agent $i$ runs a no-regret algorithm over their mixed strategy set $\mathcal{X} = \Delta(m_i)$. For this special case, the most common algorithm is the *multiplicative weights update* (MWU) algorithm (*e.g.*, Freund & Schapire (1999)), which sets $\boldsymbol{x}^t \propto \exp\big(\eta \sum_{\tau < t} \boldsymbol{u}^\tau\big)$, where $\eta = \sqrt{\log(m_i)/T}$ is an appropriately-chosen step size and $\exp(\cdot)$ is the element-wise exponential function. Multiplicative weights achieves regret bound $R(T) \lesssim \sqrt{T \log m_i}$.

## 3 OUR PROBLEM: LEARNING FROM LEARNING AGENTS

We study a setting where the principal does not initially know anything about the agents' utility functions $U_i$ except for boundedness. The principal knows the number of agents $n$ and their action sets $A_i$, oversees the agents playing the game repeatedly over $T$ rounds, and can provide *payments* and *signals* to influence the agents' behavior, in order to learn the utility functions of the agents. More formally, in each round $t = 1, \ldots T$, the following events happen in order:

1. The principal selects *payment function* $P_i^t : A_i \to [0, B]$ for each agent $i$, where $B$ is a large constant.[1] The payment $P_i^t(a_i)$ is added to agent $i$'s reward, creating a new game $\Gamma^t$ with utility functions given by $U_i^t(a) := U_i(a) + P_i^t(a_i)$.

2. The principal sends a *signal* $s_i^t \in S_i$ to each agent $i$.

3. Observing $s_i^t$ (but not $P_i^t$), each agent $i$ selects a mixed strategy $\boldsymbol{x}_i^t \in \Delta(A_i)$.

4. The principal observes the joint mixed strategy $\boldsymbol{x}^t$. Each agent $i$ gets reward $U_i^t(\boldsymbol{x}^t)$.

**Agents' contextual no-regret learning** We assume that each agent selects $\boldsymbol{x}_i^t$ using a no-regret learning algorithm, or more precisely, a *contextual* no-regret learning algorithm where signals are contexts and agents achieve no regret given any signal/context. Formally, there is a (possibly game-dependent) constant $C \leq \mathsf{poly}(M)$ such that, for every agent $i$, every signal $s_i \in S_i$, every time step $t \leq T$,[2]

$$\hat{R}_i(t, s_i) := \max_{a_i \in A_i} \sum_{\tau \leq t : s_i^\tau = s_i} \left[ U_i^\tau(a_i, \boldsymbol{x}_{-i}^\tau) - U_i^\tau(\boldsymbol{x}^\tau) \right] \leq C\sqrt{T}. \tag{1}$$

One way to achieve this guarantee is for each agent to run $|S_i|$ independent no-regret algorithms, one for each signal. However, we do not restrict to any specific algorithms: agents can run *any* algorithms satisfying the above no-regret property, such as projected gradient ascent, MWU, Exp3, and so on. In particular, agents can run full-feedback or bandit-feedback no-regret algorithms; our results are unaffected. Indeed, the agents' learning algorithms need not even be independent; they could choose their actions using a centralized algorithm. As typical no-regret algorithms do not require knowledge of the utility functions (they operate on the feedback received after each round), the agents can also be initially ignorant of their own utility functions $U_i$, just as the principal is.

---

[1] All our positive results will only require $B = 2$, since the utility functions are bounded by 1.

[2] *Anytime* no-regret is not a strong requirement. We show in Appendix B.1 that any algorithm with the usual no-regret guarantee under adversarial environments automatically satisfies anytime no-regret.

**Mixed strategies and agent randomization**   Our model stipulates that the principal observes the full joint mixed strategy $\boldsymbol{x}^t$, instead of a sampled pure strategy profile $\boldsymbol{a}^t \sim \boldsymbol{x}^t$. This stipulation, however, is not at all vital and could be removed with only minimal effect on the results. In particular, suppose that each agent $i$, instead of announcing a mixed strategy $\boldsymbol{x}_i^t$ in each round, samples an action $a_i^t \sim \boldsymbol{x}_i^t$ and announces the "mixed strategy" that assigns all mass to $a_i^t$. Then the principal only observes $a_i^t$. The difference now is that, since the agents are randomizing, the regret bound (1) cannot hold *deterministically*; there will be some stochastic approximation error term that can be bounded by Azuma-Hoeffding inequality, so (1) only holds *with high probability*. In that setting, one can think of the results of this paper as conditional on the high-probability event that (1) holds.

In particular, lower bounds that apply to the setting where the principal only observes the realized action profile also apply to the setting where the principal observes the mixed strategy profile. We will use this fact freely to prove lower bounds, in Theorem 4.4.

**Our goal**   We aim to design algorithms for the principal to learn the agents' utility functions $U_i$ by designing the payment functions $P_i^t$ and signals $s_i^t$, for any no-regret learning agents. However, this goal as currently stated is impossible, because agents' regrets are only affected by the *utility differences* between actions, namely, $U_i(a_i, \boldsymbol{a}_{-i}) - U_i(a_i', \boldsymbol{a}_{-i})$, and the behaviors of typical no-regret algorithms (such as MWU) only depend on those differences. In other words, if we create another game $\tilde{\Gamma}$ with $\tilde{U}(a_i, \boldsymbol{a}_{-i}) = U_i(a_i, \boldsymbol{a}_{-i}) + W_i(\boldsymbol{a}_{-i})$ for all $\boldsymbol{a} \in A$, where $W_i : A_{-i} \to \mathbb{R}$ is an arbitrary function not depending on agent $i$'s action, there is no way to distinguish $\Gamma$ from $\tilde{\Gamma}$ using only agents' behavioral data – that is, games $\Gamma$ and $\tilde{\Gamma}$ are *strategically equivalent*. Thus, we can only determine utility functions *up to* strategic equivalence.

Formally, given a game $\Gamma$ and precision $\varepsilon$, we say that the principal's algorithm $\varepsilon$-*learns* the game $\Gamma$ if it outputs utility functions $\tilde{U}_i : A \to \mathbb{R}$ such that there exist functions $W_i : A_{-i} \to \mathbb{R}$ satisfying

$$\left| U_i(\boldsymbol{a}) + W_i(\boldsymbol{a}_{-i}) - \tilde{U}_i(\boldsymbol{a}) \right| \leq \varepsilon, \quad \forall i \in [n], \ \forall \boldsymbol{a} \in A. \tag{2}$$

The goal of the principal is to $\varepsilon$-learn $\Gamma$ in as few rounds as possible. Learn a game up to strategic equivalence is sufficient for most practical purposes. For example, the sets of $\varepsilon$-Nash equilibria and $\varepsilon$-correlated equilibria are invariant under strategic equivalence. Learning under strategic equivalence also enables *steering* the learners toward certain outcomes, an application in Section 5.

While our paper focuses on the problem of minimizing the number of *rounds* it takes to learn the game, an alternative goal might be to minimize the total payment to do so. However, we show in Section A that the minimal achievable payment is upper-bounded and lower-bounded by the minimal number of rounds up to constant factors, so these two problems are quantitatively similar.

## 4   LEARNING A GAME BY PAYING NO-REGRET LEARNERS

We design efficient algorithms for the principal to $\varepsilon$-learn a game by paying no-regret learning agents. We will start with the single-agent case to illustrate the main ideas of our algorithms, before proceeding to the multi-agent case.

### 4.1   THE SINGLE-AGENT CASE

In the single-case case ($n = 1$), it is more convenient to view the single agent's utility function as a vector $\boldsymbol{u} \in [0, 1]^m$, and similarly the payment $P^t : [m] \to \mathbb{R}$ as vector $\boldsymbol{p}^t \in \mathbb{R}^m$ and total utility $\boldsymbol{u}^t := \boldsymbol{u} + \boldsymbol{p}^t$. To simplify notations, we subtract the average utility of all actions from the utility of each action: $\boldsymbol{u} \leftarrow \boldsymbol{u} - \frac{\langle \mathbf{1}, \boldsymbol{u} \rangle}{m} \mathbf{1}$, so that $\boldsymbol{u} \in [-1, 1]^m$ and $\langle \mathbf{1}, \boldsymbol{u} \rangle = 0$. By the discussion before (2), this does not change the principal's learning problem. Our algorithm will not need signaling in the single-agent case, so it will be enough to set $|S_i| = 1$.

The main challenge of learning a game from a no-regret agent is the history-dependency of the agent's behavior. To fix ideas, suppose the agents has two actions $A$ and $B$ with unknown utility gap $u_A - u_B = \Delta > 0$. A straightforward attempt to learn the gap $\Delta$ is to try different payments for action $B$ (while paying 0 for action $A$) in a binary-search manner: the payment at which the agent just starts to play action $B$ should reveal the value of $\Delta$. However, that approach does not work for

a no-regret learning agent because no-regret algorithms may not respond instantaneously to changes to the payment function. Historical payments affect the agent's future behavior. Moreover, the agent may incur *negative* regret over time, making it difficult to learn anything from the agent's behavior on subsequent rounds. For example, if an agent has regret $-K$ for all actions, then one cannot say anything at all about how the agent will behave in the next $K$ rounds.

The key idea of our algorithm is to *imagine the principal and the agent as playing a zero-sum game* where the principal selects the payment function $p$ from some set $\mathcal{P}$ to be specified later, the agent selects mixed strategy $x \in \Delta(m)$, the agent's utility is given by $\langle u + p, x \rangle$, and the principal's utility is $-\langle u + p, x \rangle$. Call this game $\Gamma_0$. In particular, by setting $\mathcal{P} = \{p \in [0,2]^m : \langle \mathbf{1}, p \rangle = m\}$, the zero-sum game $\Gamma_0$ has the following property:

**Lemma 4.1.** *In the zero-sum game $\Gamma_0$, every $\varepsilon$-Nash equilibrium strategy for the principal has the form $p = \mathbf{1} - u + z$, where $\|z\|_1 \le 4m\varepsilon$.*

*Proof.* Setting $p = \mathbf{1} - u$ guarantees $\langle u + p, x \rangle = \langle \mathbf{1}, x \rangle = 1$ for every $x \in \Delta(m)$. Thus, in every $\varepsilon$-Nash equilibrium, the agent's utility is at most $1 + \varepsilon$. Now suppose for contradiction that $(p, x)$ is an $\varepsilon$-Nash equilibrium with $\|p + u - \mathbf{1}\|_1 > 4m\varepsilon$. Then since $\langle p + u - \mathbf{1}, \mathbf{1} \rangle = 0$ by construction, there must be an action $a$ for which $(p + u - \mathbf{1})[a] > 2\varepsilon$, *i.e.*, $(u + p)[a] > 1 + 2\varepsilon$. But then the agent has an $\varepsilon$-profitable deviation to action $a$. $\square$

It is well known that no-regret learning algorithms converge on average to Nash equilibria in zero-sum games. In particular, if both principal and agent run no-regret algorithms, and $R_0$ is the regret after $T$ timesteps for the principal, then the average principal strategy $\frac{1}{T}\sum_{t=1}^{T} p^t$ is an $\varepsilon$-Nash equilibrium for $\varepsilon \lesssim (R_0 + C\sqrt{T})/T$. Here, we use the projected gradient descent algorithm for the principal. Note that, although the principal's utility function $p \mapsto -\langle u + p, x \rangle$ depends on $u$ (which the principal does not know), the gradient of the principal's utility function is $-x$, which does not depend on $u$. Thus, the principal can run projected gradient descent without the knowledge of $u$. The resulting algorithm is formalized in Algorithm 1.

---

**Algorithm 1** Principal's learning algorithm for a single no-regret agent

1: $p^1 \leftarrow \mathbf{1}$
2: **for** each time $t = 1, \dots, T$ **do**
3:      principal selects payment vector $p^t \in \mathcal{P}$, observes strategy $x^t$ played by the agent
4:      principal sets $p^{t+1} \leftarrow \Pi_{\mathcal{P}}[p^t - \eta x^t]$     $\triangleright \eta = \sqrt{m/T}$ is the step size
5: **return** $-\frac{1}{T}\sum_{t=1}^{T} p^t$

---

**Theorem 4.2.** *Algorithm 1 $\varepsilon$-learns any single-agent game $\Gamma$ in $T = \mathcal{O}(\frac{m^3 + C^2 m^2}{\varepsilon^2})$ rounds.*

*Proof.* Let $\bar{p} = \frac{1}{T}\sum_{t=1}^{T} p^t$ be the average payment. From the preliminaries, the regret bound of the principal is given by $R_0 \le BG\sqrt{T}$ where $B \lesssim \sqrt{m}$ and $G = 1$. By the previous paragraph, the average play between the principal and the agent is an $\frac{R_0 + C\sqrt{T}}{T}$-Nash equilibrium. Then, by Theorem 4.1, we have

$$\varepsilon = \|\bar{p} + u - \mathbf{1}\|_\infty \le \|\bar{p} + u - \mathbf{1}\|_1 \lesssim 4m\frac{R_0 + C\sqrt{T}}{T} \lesssim \frac{m}{\sqrt{T}}(C + \sqrt{m}).$$

Solving for $T$ yields the desired result. $\square$

The zero-sum-game idea of Algorithm 1 is surprisingly simple and powerful. The principal moves the payment vector in the opposite direction of the agent's mixed strategy every round, and the negative average payment vector ultimately becomes an accurate estimate of the agent's utility vector. This idea works for any no-regret learning algorithm of the agent.

Another possible method to overcome history-dependency is to use signals: whenever the payment vector is changed, send a new signal to the agent to disentangle from the history. Signaling allows us to implement a binary-search algorithm to learn the game. However, that would require $m \log(1/\varepsilon)$ signals, one for each step of the binary search, and the total number of rounds would be at least

$C^2 m/\varepsilon^2 \cdot \log(1/\varepsilon)$, whereas our Algorithm 1 achieves the better dependence of $1/\varepsilon^2$, saving a logarithmic term, without using signals.

## 4.2 THE MULTI-AGENT CASE

We then consider the multi-agent case. Our algorithm for the multi-agent case will combine the single-agent algorithm with signaling. Intuitively, our algorithm uses signals to induce the action profile $\boldsymbol{a}_{-i}$ among other agents without increasing their regret by too much. More precisely, we set the signal set as $S_i := A_i \cup \{\bot\}$ where $\bot$ is a special signal indicating that $i$'s utility is the one being learned at the moment. For every action profile $\boldsymbol{a}_{-i} \in A_{-i}$, we send signal $\bot$ to agent $i$ and the desired action $a_j$ for each agent $j \neq i$ to learn $U_i(\cdot, \boldsymbol{a}_{-i})$. This idea is formalized in Algorithm 2.

---

**Algorithm 2** Principal's learning algorithm for multiple no-regret agents

1: $t \leftarrow 1$
2: **for** every agent $i = 1, \ldots, n$ **do**
3:      **for** every action profile $\bar{\boldsymbol{a}}_{-i} \in A_{-i}$ **do**
4:          $\boldsymbol{p}^1 \leftarrow \mathbf{1} \in \mathbb{R}^{A_i}$
5:          **for** timestep $\ell = 1, \ldots, L$ **do**
6:              principal sets $P_i^t(\cdot) = \boldsymbol{p}^\ell[\cdot]$ and $P_j^t(a_j) = 2\mathbb{I}\{a_j = \bar{a}_j\}$ for every $j \neq i$
7:              principal sends signals $s_i^t = \bot$ and $s_j^t = \bar{a}_j$ for every $j \neq i$
8:              principal observes profile $\boldsymbol{x}^t$ played by agents
9:              principal sets $\boldsymbol{p}^{\ell+1} \leftarrow \Pi_{\mathcal{P}}\left[\boldsymbol{p}^\ell - \eta \boldsymbol{x}_i^t\right]$     $\triangleright \eta = \sqrt{m_i/L}$ is the step size
10:              $t \leftarrow t + 1$
11:          $\tilde{U}_i(\cdot, \bar{\boldsymbol{a}}_{-i}) \leftarrow -\frac{1}{L}\sum_{\ell=1}^{L} \boldsymbol{p}^\ell$
12: **return** $\tilde{U}$

---

**Theorem 4.3.** *For some choice of parameter L, Algorithm 2 $\varepsilon$-learns any game in $\frac{\mathsf{poly}(M,C)}{\varepsilon^2}$ rounds.*

*Proof Sketch.* Since the principal always gives a large reward to agent who obey signals other than $\bot$, agent will almost always obey such signals. Thus, agents other than agent $i$ will almost always play profile $\boldsymbol{a}_{-i}$. This allows the principal to learn $U_i(\cdot, \bar{\boldsymbol{a}}_{-i})$ using the one-player algorithm from Theorem 4.2. The formal proof is deferred to Section C. $\square$

Signals are vital to this analysis. Without them, it would be possible for players to incur large *negative* regret, which harms the learning process because it allows the players to "delay" the learning until their regrets once again become non-negative. For example, if we were to execute our algorithm without signals, then by the time $\underline{T}_n(0)$ at which the outer loop reaches agent $n$, agent $n$ could have $\Omega(\overline{T}_n(0))$ regret for *every* action, making it impossible to say anything about how agent $n$ will act for the next $\Omega(\overline{T}_n(0))$ rounds. Using signals allows us to separate out the regret of agent $n$ in previous rounds from the regret of agent $n$ when its own utility function is being learned.

## 4.3 LOWER BOUND

We now turn to lower bounds. In particular, we show a lower bound that matches Theorem 4.3 up to the exponent on $M$.

**Theorem 4.4.** *In the no-regret model, any algorithm that $\varepsilon$-learns a game must take at least $\max\{\tilde{\Omega}(nM) \cdot \log\frac{1}{\varepsilon}, \frac{C^2}{4\varepsilon^2}\}$ rounds.*

*Proof sketch.* If every agent plays a pure action at each round, then the principal can only observe $\log(M)$ bits of information at each round. Learning the game requires $\Omega(nM) \cdot \log\frac{1}{\varepsilon}$ bits of information, so we need at least $\tilde{\Omega}(nM) \cdot \log\frac{1}{\varepsilon}$ rounds in total. On the other hand, to $\varepsilon$-learn the game from agents' behavior, the agents' time-average regret $\frac{C}{\sqrt{T}}$ must be smaller than $2\varepsilon$, so $T$ is at least $\frac{C^2}{4\varepsilon^2}$. The full proof is in Appendix C.2. $\square$

Theorem 4.4 shows that it is impossible to exponentially improve the dependence on any of the parameters in Theorem 4.2. For example, it implies that there can be no algorithm taking $C^2/\varepsilon^2 \cdot M^{1-\Omega(1)}$ rounds, because that would contradict the lower bound for constant $\varepsilon$ and sufficiently large $M$. We leave it as an interesting open question to close the polynomial gaps between the lower and upper bounds presented here.

# 5 STEERING NO-REGRET LEARNERS BY LEARNING THE GAME

A main motivating application of our result is the problem of *steering* no-regret learners to desirable outcomes, introduced by Zhang et al. (2024) who assume that the principal knows the game. In this section, we explore the steering problem with unknown agent utilities.

## 5.1 CORRELATED SIGNALS AND PAYMENTS

In this section, we make two modifications to our model in Section 3: (1) we allow the signals to be *correlated*; (2) we allow payments to each agent $i$ to depend not only on agent $i$'s action, but also on the signals and actions of other players. These two assumptions are proven to be necessary for the steering problem by Zhang et al. (2024). Formally, each agent has a finite signal set $S_i$. As with actions, we will write $S = S_1 \times \cdots \times S_n$ for the joint signal space. On each round $t$, the principal first commits to both a signal distribution $\mu^t \in \Delta(S)$ and a payment function $P_i^t : S \times A \to [0, B]$. The agents then select their strategies, which are functions $\phi_i^t : S_i \to \Delta(A_i)$. Then, the principal draws the joint signal $\boldsymbol{s}^t = (s_1^t, \ldots, s_n^t) \sim \mu^t$, and each agent plays $\boldsymbol{x}_i^t = \phi_i^t(s_i^t)$. As before, we assume that agents have no regret for each signal: for every signal $s_i \in S_i$,

$$\hat{R}_i(t, s_i) := \max_{a_i \in A_i} \sum_{\tau \leq t} \sum_{\boldsymbol{s}_{-i} \in S_{-i}} \mu^t(\boldsymbol{s}) \Big[ U_i^\tau(\boldsymbol{s}, a_i, \phi_{-i}^\tau(\boldsymbol{s}_{-i})) - U_i^\tau(\boldsymbol{s}, \phi^\tau(\boldsymbol{s})) \Big] \leq C\sqrt{T}.$$

where now $U_i^\tau(\boldsymbol{s}, \boldsymbol{a}) := U_i(\boldsymbol{a}) + P_i^\tau(\boldsymbol{s}, \boldsymbol{a})$.

We remark that this correlated model gives strictly more power to the principal than the previous model: if we restrict the principal to setting $\mu^t$ to be a deterministic distribution and $P_i^t$ to be dependent on agent $i$'s action $a_i$ only, the two models coincide. Therefore, the previous positive results, particularly Theorem 4.3, apply to this model as well.

The reason for the difference in the models is that the correlated signaling model makes clear in what formal sense the signals are *private*: the agents' strategies $\phi_i^t$ can only depend on $s_i$, not other agents' signals. This will allow us to steer to *correlated* equilibria.

## 5.2 WHAT OUTCOME SHOULD WE STEER TO?

The steering problem, as defined by Zhang et al. (2024), stipulates for their main results that the principal knows in advance, or be able to compute, the desired outcome that we wish to induce. Of course, in our setting, such a stipulation is unreasonable: the principal does not initially know the agents' utilities in the game $\Gamma$, so it cannot know what outcome it wishes to induce. We thus take a more direct approach: we try to maximize the average reward, less payments, of the principal. That is, we will assume that the principal has a utility function $U_0 : A \to \mathbb{R}$, and we will attempt to optimize the principal's objective, defined as the principal's utility minus payments:

$$F(T) := \tfrac{1}{T} \sum_{t=1}^T \mathop{\mathbb{E}}_{\boldsymbol{s}^t \sim \mu^t} \Big[ U_0(\phi^t(\boldsymbol{s}^t)) - \sum_{i=1}^n P_i^t(\boldsymbol{s}^t, \phi^t(\boldsymbol{s}^t)) \Big].$$

To analyze the above objective, we introduce a solution concept called *correlated equilibrium with payments* (CEP). A CEP is a correlated distribution of signals and payment functions that satisfies the usual incentive compatibility constraints. Formally, we have the following definition.

**Definition 5.1.** A *correlated profile with payments* is a pair $(\mu, P) \in \Delta(A) \times [0, B]^{[n] \times A \times A}$.[3] The vector $P$ consists of $n$ payment functions $P_i : A \times A \to [0, B]$, where $P_i(\boldsymbol{s}, \boldsymbol{a})$ is the payment to

---

[3]Here, we let $S_i = A_i$. This is WLOG due to a *revelation principle* argument (Section D.1).

agent $i$ given joint signal $s \in A$ and joint action $a \in A$. Given $(\mu, P)$, the *objective value* for the principal is defined as

$$F(\mu, P) := \mathop{\mathbb{E}}_{a \sim \mu} \Big[ U_0(a) - \sum\nolimits_{i=1}^{n} P_i(a, a) \Big],$$

An $\varepsilon$-*correlated equilibrium with payments* ($\varepsilon$-CEP) is a pair $(\mu, P)$ satisfying the incentive compatibility (IC) constraints: for every agent $i \in [n]$ and deviation function $\phi_i : A_i \to A_i$,

$$\mathop{\mathbb{E}}_{a \sim \mu} \big[ U_i^P(a, \phi_i(a_i), a_{-i}) - U_i^P(a, a) \big] \leq \varepsilon,$$

where $U_i^P(s, a) := U_i(a) + P_i(s, a)$. An 0-CEP is called a CEP.

Let $F^*(\Gamma)$ be the principal's objective value under an *optimal* CEP of game $\Gamma$:

$$F^*(\Gamma) \;=\; \max_{(\mu, P):\text{ a CEP of game } \Gamma} F(\mu, P).$$

We show that $F^*(\Gamma)$ is an upper bound on the maximum value attainable by a principal in our learning model. Relatedly, Deng et al. (2019); Lin & Chen (2025) show that a principal cannot achieve more than the Stackelberg equilibrium objective against a single no-regret agent in games with no payment. Our result generalizes to multiple no-regret agents and games with payment. The proof of Theorem 5.2 is in Appendix D.3.

**Theorem 5.2.** *Let $\Gamma$ be any game, and suppose the signal sets have size $|S_i| \leq \mathsf{poly}(m)$. Then there exist uncoupled no-regret learning algorithms for the agents such that, for any principal algorithm, the principal's objective value $F(T)$ is bounded above by $F^*(\Gamma) + o(1/\sqrt{T})$.*

### 5.3 STEERING TO OPTIMAL CEP

We now show that the principal *can* achieve the optimal CEP objective $F^*(\Gamma)$ in the limit $T \to \infty$. The algorithm (Algorithm 3) works in two stages. In the first stage, the principal uses Algorithm 2 to learn the utility functions of the agents. Then, the principal computes an optimal CEP and steers the agents to it. The steering algorithm is adapted from Zhang et al. (2024), and presented in full here for the sake of self-containment. Notably, since the principal learns the game up to an error $\varepsilon > 0$, it must give extra payments of at least $2\varepsilon$ to ensure that agents do not deviate. Theorem 5.3 shows that the principal can learn to steer agents to achieve the optimal objective $F^*(\Gamma)$ at a rate of $\mathsf{poly}(M, C)/T^{1/4}$. The proof is given in Appendix D.4.

---

**Algorithm 3** Principal's algorithm for steering without prior knowledge of utilities

1: using Algorithm 2, estimate the utility functions to precision $\varepsilon$
2: compute an optimal CEP $(\tilde{\mu}^*, \tilde{P}^*)$ of the estimated game $\tilde{\Gamma}$
3: **for** remaining rounds **do**
4:     set $\mu^t = \mu^*$ and $P_i^t(s, a) = \begin{cases} \tilde{P}_i^*(a, a) + 2\varepsilon + \rho & \text{if} \quad s = a \\ 2 & \text{if} \quad s_i = a_i, s_{-i} \neq a_{-i}. \\ 0 & \text{otherwise} \end{cases}$

---

**Theorem 5.3.** *For appropriate choices of parameters $L$ (from Algorithm 2) and $\rho$, Algorithm 3 guarantees principal objective $F(T) \geq F^*(\Gamma) - \mathsf{poly}(M, C)/T^{1/4}$ on average in $T$ rounds.*

## 6 CONCLUSIONS AND FUTURE DIRECTIONS

We showed that a principal can efficiently learn the utility functions of agents in games through payments and signals, and applied our algorithms to achieve optimal steering without prior knowledge of the game. We gave upper and lower bounds for both problems. Our results apply to arbitrary no-regret agents. We leave a few directions for future research:

- We did not optimize the polynomial dependencies on $M$, so our upper and lower bounds are off by $\mathsf{poly}(M)$ factors. We leave it as an interesting open problem to close these gaps.

- Our techniques are specialized to normal-form games, and require, for example, that the principal observe the strategy $x^t$ of the agents at every timestep. This may no longer be a reasonable assumption in, *e.g.*, *extensive-form games*, where one may wish instead to assume that we only observe *on-path* agent actions. We leave it as future work to extend our results to such settings.

- While our single-agent utility-learning algorithm only uses payments, our mutli-agent algorithm additionally uses signals. The limit of learning agents' utility functions by payments only, without using signals, is worth exploring.

- Similarly worth exploring is the problem of steering without utility learning. We proved that agents' utility functions require $\tilde{\Omega}(nM) \log \frac{1}{\varepsilon}$ rounds to learn. Can we avoid this bottleneck by steering agents to desirable outcomes without learning their entire utility functions?

### ACKNOWLEDGEMENTS

B.H.Z. was supported by the CMU Computer Science Department Hans Berliner PhD Student Fellowship. T.L. was supported by a Siebel Scholarship. Y.C. is supported by National Science Foundation grant IIS-2147187 and by Amazon. T.S. is supported by the Vannevar Bush Faculty Fellowship ONR N00014-23-1-2876, National Science Foundation grants RI-2312342 and RI-1901403, ARO award W911NF2210266, and NIH award A240108S001.

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

# A  MINIMIZING PAYMENT

Most of our paper focuses on $\varepsilon$-learning a game in as few rounds as possible. This section discusses an alternative goal: $\varepsilon$-learning a game while minimizing the total payment to the agents. We show that the minimal achievable payment is upper-bounded and lower-bounded by the number of rounds up to constant factors, so the payment minimization problem is quantitatively similar to round complexity minimization.

Formally, we define the *payment complexity* $\mathrm{PC}(n, \varepsilon)$ to be the minimal total payment required to $\varepsilon$-learn a game with $n$ agents (in the worst case over all games), and the *round complexity* $\mathrm{RC}(n, \varepsilon)$ to be the minimal number of rounds to do so. Then, we have:

**Proposition A.1.** *There exist games and no-regret agents such that*

$$\Omega\big(\mathrm{RC}(n-1, \varepsilon)\big) \leq \mathrm{PC}(n, \varepsilon) \leq \mathcal{O}\big(n \cdot \mathrm{RC}(n, \varepsilon)\big).$$

*Proof.* The inequality $\mathrm{PC}(n, \varepsilon) \leq \mathcal{O}\big(n \cdot \mathrm{RC}(n, \varepsilon)\big)$ is straightforward because the payment to each agent is bounded by $O(1)$ at each round.

To prove $\Omega\big(\mathrm{RC}(n-1, \varepsilon)\big) \leq \mathrm{PC}(n, \varepsilon)$, we reduce the utility learning problem with $n - 1$ agents to the problem with $n$ agents. Let $\Gamma_{n-1}$ be an $(n-1)$-agent game with utility functions $U_1, \ldots, U_{n-1}$. Consider an $n$-agent game $\Gamma_n$ where the $n$-th agent has two actions $A_n = \{0, 1\}$. If the $n$-th agent takes 1, then the first $n - 1$ agents all obtain utility 0 regardless of their actions; if the $n$-th agent takes 0, then the first $n - 1$ agents have the same utility functions as in game $\Gamma_{n-1}$. Further assume that the $n$-th agent's utility depends on his own action only, and in particular, $U_n(a_n = 0) = 0$ and $U_n(a_n = 1) = 1$, so the $n$-th agent takes action 1 by default. Intuitively, in order to learn the utility functions of game $\Gamma_n$, we have to incentivize the $n$-th agent to play action 0 by paying him 1 at each round, so that we can learn the first $n - 1$ agents' utility functions. This means that the total payment is lower bounded by the number of rounds to learn the $(n - 1)$-agent game.

Formally, let ALG be an algorithm for learning $\Gamma_n$ with payment complexity $\mathrm{PC}(n, \varepsilon)$. We use ALG to construct an algorithm to learn $\Gamma_{n-1}$ as follows:

- At each round $t$, do:
    - Obtain payment functions $P_1^t, \ldots, P_{n-1}^t, P_n^t$ from ALG.
    - If $P_n^t(a_n = 0) < P_n^t(a_n = 1) + 1$, then let $a_n^t = 1$ and $a_i^t = \mathrm{argmax}_{a_i \in A_i} P_i^t(a_i)$ for $i \in \{1, \ldots, n-1\}$. Return the action profile $(a_{-n}^t, a_n^t)$ to ALG.
    - If $P_n^t(a_n = 0) \geq P_n^t(a_n = 1) + 1$, then let $a_n^t = 0$ and send the payment functions $P_1^t, \ldots, P_{n-1}^t$ to the first $n - 1$ agents. Observe their actions $a_{-n}^t$ in game $\Gamma_{n-1}$. Return $(a_{-n}^t, a_n^t)$ to ALG.

- Obtain utility functions $\tilde{U}_1, \ldots, \tilde{U}_n$ from ALG. Output $\tilde{U}_1(\cdot, a_n = 0), \ldots, \tilde{U}_{n-1}(\cdot, a_n = 0)$.

By definition, if ALG outputs $\tilde{U}_1, \ldots, \tilde{U}_n$ that are $\varepsilon$-close to the utility functions of $\Gamma_n$, then the outputs $\tilde{U}_1(\cdot, a_n = 0), \ldots, \tilde{U}_{n-1}(\cdot, a_n = 0)$ are $\varepsilon$-close to the utility functions $U_1, \ldots, U_{n-1}$ of $\Gamma_{n-1}$. Each time "$P_n^t(a_n = 0) \geq P_n^t(a_n = 1) + 1$" happens, agent $n$ takes action $a_n^t = 0$, we pay at least 1, and we interact with the first $n - 1$ agents once. So, the total payment is at least $\mathrm{PC}(n, \varepsilon) \geq \mathrm{RC}(n-1, \varepsilon)$. $\square$

# B  DETAILS OMITTED FROM SECTION 3

## B.1  ANYTIME NO REGRET

Our condition on no-regret learning is that, for every signal $s_i$ (here omitted as a superscript for notational clarity), the regret $R_i(t)$ is bounded by $C\sqrt{T}$ for *every* timestep $t \leq T$, not just at $t = T$ as is conventionally required by adversarial no-regret algorithms. This is not a significantly stronger requirement:

**Proposition B.1.** *If a (possibly randomized) adversarial no-regret algorithm satisfies $R_i(T) \leq B$ with probability $1 - \delta$ against any adversary, then with probability $1 - \delta$ it also satisfies $R_i(t) \leq B$ simultaneously for all $t \leq T$ against any adversary.*

*Proof.* Suppose not, *i.e.*, suppose that there is some adversary $\mathcal{A}$ such that, with probability $> \delta$, there exists some $t \leq T$ for which $R_i(t) > B$. Then consider the adversary $\mathcal{A}'$ that acts as follows. At every time $t$, if $R_i(t-1) \leq B$, it copies $\mathcal{A}$. Otherwise, it outputs $\boldsymbol{u}^t = \boldsymbol{0}$ for all remaining timesteps. In the latter case, which by definition occurs with probability $> \delta$, adversary $\mathcal{A}'$ will also achieve $R_i(T) > B$. $\square$

# C  DETAILS OMITTED FROM SECTION 4

## C.1  PROOF OF THEOREM 4.3

As in Theorem 4.2, we will assume without loss of generality that $\sum_{a_i \in A_i} U_i(a_i, \boldsymbol{a}_{-i}) = 0$ for all agents $i$ and opponent profiles $\boldsymbol{a}_{-i}$.

We first claim that, for any agent $i$ and any given signal $s_i^t = a_i \in A_i$, the total probability mass that $i$ plays actions *other than* $a_i$ is bounded by $C\sqrt{T}$. To see this, note that whenever the principal sends signal $a_i$, the payment is always set such that $U_i^t(a_i, \boldsymbol{a}_{-i}) \geq 1 + U_i^t(a_i', \boldsymbol{a}_{-i})$. Thus, the number of times $i$ does not play $a_i^t = a_i$ quantity lower-bounds the regret $\hat{R}(T, a_i)$. The claim follows from the regret guarantee $\hat{R}(T, a_i) \leq C\sqrt{T}$.

We will refer to the iterations of the inner loop over action profiles $\boldsymbol{a}_{-i}$ as *phases*. Fix an agent $i$, and number the phases for that agent using integers $k \in \{1, \cdots, M_i = \prod_{j \neq i} m_j\}$, corresponding to strategy profiles $\bar{\boldsymbol{a}}_{-i}^1, \dots, \bar{\boldsymbol{a}}_{-i}^{M_i} \in A_{-i}$. Let $\mathcal{T}_i(k) = \{\underline{T}_i(k), \dots, \overline{T}_i(k)\}$ be the set of timesteps in agent $i$'s $k$th phase. The length of each phase is $|\mathcal{T}_i(k)| = L$. Let $B_K$ be the total probability mass placed by all agents $j \neq i$ on strategy profiles other than $\bar{\boldsymbol{a}}_{-i}^k$ throughout phases $1, \dots, K$. By the previous claim, we have

$$B_K := \sum_{k \leq K, t \in \mathcal{T}_i(k)} \left(1 - \prod_{j \neq i} \boldsymbol{x}_j^t(\bar{a}_j^k)\right) \leq \sum_{k \leq K, t \in \mathcal{T}_i(k), j \neq i} \left(1 - \boldsymbol{x}_j^t(\bar{a}_j^k)\right) \leq nmC\sqrt{T}.$$

By the principal's regret bound in each phase, we must have

$$\sum_{t \in \mathcal{T}_i(k)} U_i^t(\boldsymbol{x}_i^t, \bar{\boldsymbol{a}}_{-i}^k) = \sum_{t \in \mathcal{T}_i(k)} U_i(\boldsymbol{x}_i^t, \bar{\boldsymbol{a}}_{-i}^k) + \sum_{t \in \mathcal{T}_i(k)} P_i^t(\boldsymbol{x}_i^t)$$

$$\leq \sum_{t \in \mathcal{T}_i(k)} U_i(\boldsymbol{x}_i^t, \bar{\boldsymbol{a}}_{-i}^k) + \sum_{t \in \mathcal{T}_i(k)} \left[1 - U_i(\boldsymbol{x}_i^t, \bar{\boldsymbol{a}}_{-i}^k)\right] + R_0$$

$$= L + R_0$$

where the inequality follows from the facts that 1) the principal's regret is bounded, 2) $P_i^t(\cdot) = 1 - U_i(\cdot, \bar{\boldsymbol{a}}_{-i}^k)$ is a valid unilateral deviation for the principal.

Fix some $K \leq M_i$ and $a_i \in A_i$. By the anytime regret bound of agent $i$ under signal $\perp$, we have

$$\sum_{k \leq K, t \in \mathcal{T}_i(k)} U_i^t(a_i, \boldsymbol{x}_{-i}^t) \leq \sum_{k \leq K, t \in \mathcal{T}_i(k)} U_i^t(\boldsymbol{x}_i^t, \boldsymbol{x}_{-i}^t) + \hat{R}_i(T_i(k), \perp)$$

$$\leq 2B_K + \sum_{k \leq K, t \in \mathcal{T}_i(k)} U_i^t(\boldsymbol{x}_i^t, \bar{\boldsymbol{a}}_{-i}^k) + \hat{R}_i(T_i(k), \perp)$$

$$\leq 2nmC\sqrt{T} + K(L + R_0) + nC\sqrt{T}.$$

Moving $KL$ to the left and writing $U_i^t(a_i, \boldsymbol{x}_{-i}^t)$ as $U_i(a_i, \boldsymbol{x}_{-i}^t) + P_i^t(a_i)$,

$$\sum_{k=1}^K \underbrace{\frac{1}{L} \sum_{t \in \mathcal{T}_i(k)} [U_i(a_i, \boldsymbol{x}_{-i}^t) + P_i^t(a_i) - 1]}_{\text{denoted by } \varepsilon_i(k, a_i)} \leq \frac{1}{L}\left(R_0 K + 3nmC\sqrt{T}\right).$$

The error we need to bound is $\|\varepsilon_i(k, \cdot)\|_\infty$. Since the above inequality holds for any $a_i$, and $\sum_{a_i} \varepsilon_i(k, \cdot) = 0$ by definition, it follows that

$$\left\| \sum_{k=1}^K \varepsilon_i(k, \cdot) \right\|_\infty = \max_{a_i \in A_i} \left| \sum_{k=1}^K \frac{1}{L} \sum_{t \in \mathcal{T}_i(k)} [U_i(a_i, \boldsymbol{x}_{-i}^t) + P_i^t(a_i) - 1] \right| \leq \frac{m}{L} \left( R_0 K + 3nmC\sqrt{T} \right).$$

By triangle inequality, we have

$$\|\varepsilon_i(k, \cdot)\|_\infty = \left\| \sum_{k'=1}^k \varepsilon_i(k', \cdot) - \sum_{k'=1}^{k-1} \varepsilon_i(k', \cdot) \right\|_\infty \leq \frac{2m}{L} \left( R_0 M + 3nmC\sqrt{T} \right).$$

Finally, substituting $R_0 \lesssim \sqrt{mL}$ and $T \leq nML$, we arrive at

$$\|\varepsilon_i(k, \cdot)\|_\infty \lesssim \frac{1}{\sqrt{L}} \left( m^{3/2} M + n^{3/2} m^2 C M^{1/2} \right).$$

Taking $L = \mathcal{O}(\frac{m^3 M^2 + n^3 m^4 M C^2}{\varepsilon^2})$ completes the proof, with $T \leq nML = \mathcal{O}(\frac{nm^3 M^3 + n^4 m^4 M^2 C^2}{\varepsilon^2})$.

## C.2 PROOF OF THEOREM 4.4

We prove both terms in the max separately. For the first term, suppose that $U_i(1, \cdot) = 0$ for all agents $i$, and $U_i(a_i, a_{-i}) \sim \{0, 2\varepsilon, 4\varepsilon, \ldots, 1\}$ i.i.d. for $2 \leq a \leq m_i$ and $a_{-i} \in A_{-i}$. Thus the utility $U$ is uniformly sampled from a set of $\Omega(1/\varepsilon)^K$ possible utilities, where

$$K = \sum_{i=1}^n \left( (m_i - 1) \prod_{j \neq i} m_j \right) \geq \frac{nM}{2}.$$

Each utility function differs by $2\varepsilon$, it follows that $\varepsilon$-learning a game sampled from this family entails exactly outputting the utility $U$. Suppose that, as discussed in Section 3, the no-regret algorithms always output pure strategies $a_i^t \in A_i$. Then on each round, the principal only observes a single action profile $a \in A$, which only conveys $\log M$ bits of information. Therefore, $\varepsilon$-learning the game takes at least

$$\frac{K \log(\Omega(1/\varepsilon))}{\log M} \gtrsim \frac{nM \log(1/\varepsilon)}{\log M}$$

rounds, as desired.

For the second term, suppose $\varepsilon \leq \frac{C}{2\sqrt{T}}$. Let $Z_i : A \to [0, \varepsilon]$ be any function, and suppose that every agent plays according to utility function $U_i + Z_i$ instead of $U_i$ using an algorithm with $\frac{C}{2}\sqrt{T}$ regret. Such an agent incurs at most $\frac{C}{2}\sqrt{T} + \varepsilon T \leq C\sqrt{T}$ regret with respect to $U_i$. Such an agent is completely indistinguishable from an agent who has true utility $U_i + Z_i$ and runs an algorithm with regret $C\sqrt{T}$, and therefore the principal can never distinguish between these two possibilities. Since this is true for any $Z_i$, this means that the principal cannot learn $U_i$ to accuracy better than $\varepsilon \leq \frac{C}{2\sqrt{T}}$. Thus, $T$ must be at least $\frac{C^2}{4\varepsilon^2}$ for the principal to $\varepsilon$-learn the game.

# D DETAILS OMITTED FROM SECTION 5

## D.1 REVELATION PRINCIPLE FOR CEPS

In this section, we formulate a general version of the revelation principle for CEPs.

**Definition D.1** (Non-canonical CEP). A (non-canonical, agent-form) $\epsilon$-CEP is a distribution $\pi \in \Delta(S \times \mathcal{P} \times A_1^{S_1} \times \cdots \times A_n^{S_n})$, where $\mathcal{P} = [0, B]^{[n] \times A \times A}$ is the set of payment functions, such that, for any player $i$ and any map $\psi_i : A_i \to \Delta(A_i)$, we have

$$\mathbb{E}_{(\boldsymbol{s}, P, \phi) \sim \pi} \left[ U_i^P(\boldsymbol{s}, (\psi_i \circ \phi_i)(s_i), \phi_{-i}(\boldsymbol{s}_{-i})) - U_i^P(\boldsymbol{s}, \phi(\boldsymbol{s})) \right] \leq 0.$$

The objective value is given by

$$\mathbb{E}_{(\boldsymbol{s}, P, \phi) \sim \pi} \left[ U_0(\phi(\boldsymbol{s})) - P(\boldsymbol{s}, \phi(\boldsymbol{s})) \right].$$

We say that $\pi$ is *canonical* if the payment function $P \sim \pi$ is constant, and for every player $i$, $S_i = A_i$ and $\phi_i$ is the identity map. Note that canonical CEPs are precisely the CEPs according to Theorem 5.1.

**Proposition D.2** (Revelation principle for CEPs). *Every CEP is equivalent to a canonical CEP, in the sense that, for every CEP $\pi$, there is a canonical CEP $(\mu', P')$ achieving the same principal objective value.*

*Proof.* Given a CEP $\pi$, set $\mu' \in \Delta(A)$ to be the distribution that samples $(s, \phi) \sim \pi$ and then samples and outputs $a \sim \phi(s)$. Then define $P'_i : A \times A \to [0, B]$ by

$$P'_i(a, a') = \mathop{\mathbb{E}}_{(s,P) \sim \pi|a} P_i(s, a'),$$

where $(s, P) \sim \pi|a$ denotes sampling $(s, P)$ with probability proportional to $\pi(s, P) \cdot \phi(a|s)$. Then note that, for any $\psi_i : A_i \to A_i$, we have

$$
\begin{aligned}
\mathop{\mathbb{E}}_{a \sim \mu'}[U_i^{P'}(a, \psi_i(a_i), a_{-i})] &= \mathop{\mathbb{E}}_{a \sim \mu'}[U_i(\psi_i(a_i), a_{-i}) + P'_i(a, \psi_i(a_i), a_{-i})] \\
&= \mathop{\mathbb{E}}_{\substack{a \sim \mu' \\ (s,P) \sim \pi|a}}[U_i(\psi_i(a_i), a_{-i}) + P_i(s, \psi_i(a_i), a_{-i})] \\
&= \mathop{\mathbb{E}}_{\substack{(s,P,\phi) \sim \mu \\ a \sim \phi(s)}}[U_i(\psi_i(a_i), a_{-i}) + P_i(s, \psi_i(a_i), a_{-i})] \\
&= \mathop{\mathbb{E}}_{(s,P,\phi) \sim \pi}\left[U_i^P(s, (\psi_i \circ \phi_i)(s_i), \phi_{-i}(s_{-i}))\right]
\end{aligned}
$$

Thus, if $\psi_i$ is a profitable deviation for agent $i$ in $(\mu', P')$, then it is also a profitable deviation in $\pi$. But the non-canonical CEP $\pi$ does not have profitable deviations, so $(\mu', P')$ is also a CEP. $\square$

### D.2 Additional Properties of CEPs

We present some additional properties of CEPs.

First, optimal CEPs can be efficiently computed when the game is known.

**Proposition D.3.** *Given a game $\Gamma$ of size $M$, an optimal CEP $(\mu^*, P^*)$ and its principal objective $F^*(\Gamma)$ can be computed in $\mathrm{poly}(M)$-time by a linear program.*

*Proof.* Define change of variables

$$Q_i(a_i) := \mu(a_i) \cdot \mathop{\mathbb{E}}_{a \sim \mu|a_i} P_i(a, a).$$

That is, $Q_i(a_i)$ is the $\mu$-weighted total payment given to agent $i$ across all strategy profiles on which agent $i$ is recommended action $a_i$. Consider the following linear program:

$$
\max_{\mu, Q_i, \varepsilon_i} \quad \sum_{a \in A} \mu(a) U_0(a) - \sum_{\substack{i \in [n] \\ a_i \in A_i}} Q_i(a_i) \quad \text{s.t.}
$$

$$
\sum_{a_{-i} \in A_{-i}} \mu(a)[U_i(a'_i, a_{-i}) - U_i(a)] - Q_i(a_i) \leq \varepsilon_i(a_i) \qquad \forall i \in [n], a_i, a'_i \in A_i
$$

$$
\sum_{a_i \in A_i} \varepsilon_i(a_i) \leq 0 \qquad \forall i \in [n] \tag{3}
$$

$$
\sum_{a \in A} \mu(a) = 1
$$

$$
0 \leq Q_i(a_i) \leq \mu(a_i) \qquad \forall i \in [n], a_i, a'_i \in A_i.
$$

This LP is equivalent to computing the optimal 0-CEP because for any feasible solution $(\mu, Q)$ of the LP, the payment functions defined by $P_i(a, a) = \frac{Q_i(a_i)}{\mu(a_i)}$ and $P_i(a, a') = 0$ if $a' \neq a$ together with $\mu$ constitute a feasible 0-CEP with the same objective value. This LP has $\mathrm{poly}(M)$ variables and constraints, so the proof is complete. $\square$

Second, we show that the assumption that the payments can be signal-dependent is not innocuous, except when the payment at equilibrium is zero. There exist games where a CEP with signal-dependent payment is strictly better than a CEP with signal-independent payment.

**Proposition D.4** (Correlation does not help when no payments are allowed in equilibrium). *The $0$-CEPs with $\mathbb{E}_{\boldsymbol{a}\sim\mu} P(\boldsymbol{a}, \boldsymbol{a}) = 0$ are exactly the correlated equilibria.*

*Proof.* In the LP (3), this is equivalent to setting $Q_i(\cdot) = 0$ for every agent $i$, in which case (3) is just the LP characterizing correlated equilibria. □

However, when the payment at equilibrium is positive, it is possible for signal-dependent payments to help the principal.

**Proposition D.5** (Signal-dependent payments can help in general). *There exists a game $\Gamma$, and principal utility function $U_0$, such that the optimal value of (3) is greater than the objective value of the optimal CEP in which $P(\boldsymbol{s}, \boldsymbol{a})$ depends on $\boldsymbol{a}$ but not $\boldsymbol{s}$.*

*Proof sketch.* In the normal-form game below, P1 and P2 play matching pennies, and the principal is willing to pay a large amount to avoid a particular pure profile.

|   | $X$ | $Y$ |
|---|---|---|
| $X$ | $-\infty, 0, 1$ | $0, 1, 0$ |
| $Y$ | $0, 1, 0$ | $0, 0, 1$ |

P1 chooses the row, P2 chooses the column. In each cell, the principal's utility is listed first, then P1's, then P2's. Now consider the following CEP: The principal mixes evenly between recommending $(X, Y)$, $(Y, X)$, and $(Y, Y)$. If the principal recommends $(Y, X)$, it also promises a payment of $1$ to P2 if P2 follows the recommendation $X$. This CEP has principal objective value $-1/3$, and no signal-independent CEP can match that value. The full proof is given in Section D.2.1. □

In the language of Monderer & Tennenholtz (2003), a CEP with $k = \mathbb{E}_{\boldsymbol{a}\sim\mu} P(\boldsymbol{a})$ is called a *$k$-implementable correlated equilibrium*.[4] They show that all correlated equilibria are $0$-implementable, but do not show the converse. Our results improve upon theirs by 1) showing the converse (Theorem D.4), and 2) analyzing the $k > 0$ case, in particular, by incorporating a principal objective and showing how to compute the optimal CEP.

### D.2.1 Complete proof of Theorem D.5

We first show that the claimed CEP is actually a CEP.

- Conditioned on P1 being recommended $X$, P2's action is deterministically $Y$, against which $X$ is the best response for P1.

- Conditioned on P1 being recommended $Y$, P2's action is uniform random, against which $Y$ is a best response for P1.

- Conditioned on P2 being recommended $X$, P1's action is deterministically $Y$, against which the principal's promised payment of $1$ makes $X$ a best response for P2.

- Conditioned on P2 being recommended $Y$, P1's action is uniform random, against which $Y$ is a best response for P2.

It remains to show that the objective value $-1/3$ cannot be achieved by any CEP in which payments are signal-independent. We prove it by contradiction. Suppose there is a CEP $(\mu, P)$ with signal-independent $P$ that achieves objective value at least $-1/3$. Note that $\mu \in \Delta(A)$ is a correlated

---

[4]Instead of our condition of *ex-interim* IC, Monderer & Tennenholtz (2003) insist on *dominant-strategy* IC, that is, they insist that $U_i^P(s, s_i, a_{-i}) \geq U_i^P(s, a)$ for *every* $s$ and $a$. However, this requirement does not change anything in equilibrium, because one can always set $P(s, s_i, a_{-i})$ when $s_i = a_i$ and $s \neq a$ to be so large that playing $a_i$ becomes dominant. Indeed, Monderer & Tennenholtz (2003) do this to establish their results on implementation; Zhang et al. (2024) do this in their steering algorithms; and we will do the same in Section 5.

equilibrium of the game with utility function $U + P$. Because the principal's utility without the payment part is always non-positive, for the objective to be at least $-1/3$, the expected payment to the two players $\sum_{\boldsymbol{a}} \mu(\boldsymbol{a}) P(\boldsymbol{a})$ cannot exceed $1/3$.

Since $U_P(X, X) = -\infty$, we must have $\mu(X, X) = 0$. We then analyze the incentive compatibility constraints for the two players:

- When P2 is recommended $X$, P2 knows that P1 is recommended $Y$ (because $(X, X)$ is not possible), so in order to ensure P2 has no incentive to deviate from $X$ to $Y$, we must have

$$U_2^P(Y, X) \geq U_2^P(Y, Y) \iff 0 + P_2(Y, X) \geq 1 + P_2(Y, Y) \implies P_2(Y, X) \geq 1.$$

  Since the expected payment is at least $\mu(Y, X) P_2(Y, X)$ but is at most $1/3$, we must have

$$\mu(Y, X) \leq 1/3.$$

- When P2 is recommended $Y$, P2 believes that the recommendation to P1 is $X$ and $Y$ with probability $\mu(X, Y)$ and $\mu(Y, Y)$, respectively, so to prevent P2 from deviation $Y \to X$, the expected utilities of P2 under actions $Y$ and $X$ should satisfy:

$$\underbrace{\mu(X, Y) \cdot \big(U_2(X, Y) + P_2(X, Y)\big) + \mu(Y, Y) \cdot (U_2(Y, Y) + P_2(Y, Y))}_{\text{P2's expected utility when taking action } Y \text{ given recommendation } Y}$$

$$\geq \underbrace{\mu(X, Y) \cdot \big(U_2(X, X) + P_2(X, X)\big) + \mu(Y, Y) \cdot \big(U_2(Y, X) + P_2(Y, X)\big)}_{\text{P2's expected utility when taking action } X \text{ given recommendation } Y}$$

$$\iff \mu(X, Y) \cdot \big(0 + P_2(X, Y)\big) + \mu(Y, Y) \cdot (1 + P_2(Y, Y))$$

$$\geq \mu(X, Y) \cdot \big(1 + P_2(X, X)\big) + \mu(Y, Y) \cdot \big(0 + P_2(Y, X)\big)$$

$$\implies \mu(X, Y) \cdot P_2(X, Y) + \mu(Y, Y) \cdot P_2(Y, Y)$$

$$\geq \mu(X, Y) \cdot \big(1 + P_2(X, X)\big) + \mu(Y, Y) \cdot P_2(Y, X) - \mu(Y, Y).$$

  Because payments are non-negative and $P_2(Y, X) \geq 1$, the above implies

$$\mu(X, Y) \cdot P_2(X, Y) + \mu(Y, Y) \cdot P_2(Y, Y)$$

$$\geq \mu(X, Y) \cdot \big(1 + P_2(X, X)\big) + \mu(Y, Y) \cdot P_2(Y, X) - \mu(Y, Y).$$

$$\geq \mu(X, Y).$$

  So, the total expected payment to P2 is at least

$$\mu(X, Y) P_2(X, Y) + \mu(Y, Y) P_2(Y, Y) + \mu(Y, X) P_2(Y, X) \tag{4}$$

$$\geq \mu(X, Y) + \mu(Y, X).$$

- When P1 is recommended $Y$, P1 knows that P2's action is $X$ with probability $\mu(Y, X)$ and $Y$ with probability $\mu(Y, Y)$, so to prevent P1 from deviation $Y \to X$, P1's expected utilities under actions $Y$ and $X$ should satisfy:

$$\underbrace{\mu(Y, X) \cdot \big(U_1(Y, X) + P_1(Y, X)\big) + \mu(Y, Y) \cdot (U_1(Y, Y) + P_1(Y, Y))}_{\text{P1's expected utility when taking action } Y \text{ given recommendation } Y}$$

$$\geq \underbrace{\mu(Y, X) \cdot \big(U_1(X, X) + P_1(X, X)\big) + \mu(Y, Y) \cdot (U_1(Y, Y) + P_1(Y, Y))}_{\text{P1's expected utility when taking action } X \text{ given recommendation } Y}$$

$$\iff \mu(Y, X) \cdot \big(1 + P_1(Y, X)\big) + \mu(Y, Y) \cdot \big(0 + P_1(Y, Y)\big)$$

$$\geq \mu(Y, X) \cdot \big(0 + P_1(X, X)\big) + \mu(Y, Y) \cdot \big(1 + P_1(Y, Y)\big).$$

  Since payments are non-negative, the above implies

$$\mu(Y, X) \cdot P_1(Y, X) + \mu(Y, Y) \cdot P_1(Y, Y) \tag{5}$$

$$\geq \mu(Y, X) \cdot P_1(X, X) + \mu(Y, Y) + \mu(Y, Y) \cdot P_1(Y, Y) - \mu(Y, X)$$

$$\geq \mu(Y, Y) - \mu(Y, X).$$

Now, adding (4) and (5), the total expected payment to the two players is at least

$$\sum_{\boldsymbol{a}} \mu(\boldsymbol{a}) P(\boldsymbol{a}) \geq \mu(X, Y) + \mu(Y, X) + \mu(Y, Y) - \mu(Y, X)$$

$$= \mu(X, Y) + \mu(Y, Y) = 1 - \mu(Y, X) \geq 2/3 > 1/3$$

because $\mu(Y, X) \leq 1/3$, which contradicts the condition that the expected payment is at most $1/3$.

## D.3 PROOF OF THEOREM 5.2

Suppose that the agents run *no contextual swap regret* algorithms. Concretely, an agent has no contextual swap regret if

$$\hat{R}_i(t, s_i) := \max_{\psi_i : A_i \to A_i} \sum_{\tau \leq t} \sum_{\boldsymbol{s}_{-i} \in S_{-i}} \mu^t(\boldsymbol{s}) \Big[ U_i^\tau(\boldsymbol{s}, (\psi_i \circ \phi_i)(s_i), \phi_{-i}^\tau(\boldsymbol{s}_{-i})) - U_i^\tau(\boldsymbol{s}, \phi^\tau(\boldsymbol{s})) \Big] \leq \varepsilon T.$$

where $\varepsilon \to 0$ as $T \to \infty$. For typical no contextual swap regret algorithms, $\varepsilon = \mathcal{O}(\frac{C}{\sqrt{T}})$ where $C$ depends on the game and the number of signals. Clearly, contextual swap regret is a stronger benchmark than the standard (external) notion of regret.

Then, after sufficiently many rounds $T$, by definition, we have that the correlated strategy profile

$$\pi := \frac{1}{T} \sum_{t=1}^{T} (\mu^t, P^t, \phi^t) \in \Delta(S \times [0, B]^{[n] \times A \times A} \times A_1^{S_1} \times \cdots \times A_n^{S_n})$$

is a non-canonical $(\varepsilon \cdot \max_i |S_i|)$-CEP in the sense of Section D.1. Therefore, by the revelation principle for CEPs (Theorem D.2), the principal objective value is bounded by that of the best $(\varepsilon \cdot \max_i |S_i|)$-CEP, which is then bounded by $F^*(\Gamma) + \mathcal{O}(\varepsilon)$.

## D.4 PROOF OF THEOREM 5.3

From the analysis of Theorem 4.3, Algorithm 2 learns a game to precision $\varepsilon = \mathsf{poly}(M, C)/\sqrt{L}$. (Note that we do not express $\varepsilon$ as a function of $T$ because $T$ is now the total number of rounds across both learning and steering stages. )

Since $\tilde{U}$ and $U$ differ by only $\varepsilon$ (up to agent-independent terms), every CEP of $\tilde{\Gamma}$ is a $2\varepsilon$-CEP of $\Gamma$. The payment function $P_i^t$ for the steering stage then ensures that, when given signal $s_i$, it is *dominant* for agent $i$ to play $a_i = s_i$. Formally, regardless of how other agents act, we have

$$U_i^t(\boldsymbol{s}, a_i, \boldsymbol{a}_{-i}) - U_i^t(\boldsymbol{s}, a_i', \boldsymbol{a}_{-i}) \geq \rho, \quad \forall \boldsymbol{s} \in A, a_i = s_i, \forall a_i' \neq s_i, \forall \boldsymbol{a}_{-i} \in A_{-i}.$$

Further, from the analysis of Theorem 4.3, agent $i$'s regret against following signals $s_i \neq \perp$ is always nonnegative. Therefore, by agent $i$'s regret bound, there are at most $C\sqrt{T}/\rho$ rounds on which agent $i$ fails to obey recommendation $s_i$ in the steering stage. By a union bound, there are therefore $mnC\sqrt{T}/\rho$ rounds in the steering stage on which $\boldsymbol{a}^t \neq \boldsymbol{s}^t$. Thus, the principal's suboptimality is bounded by

$$F^*(\Gamma) - F(T) \leq \underbrace{F^*(\Gamma) - F^*(\tilde{\Gamma})}_{(1)} + \underbrace{\frac{(2n+1)nML}{T}}_{(2)} + \underbrace{n(2\varepsilon + \rho)}_{(3)} + \underbrace{\frac{(2n+1)mnC}{\rho\sqrt{T}}}_{(4)}$$

$$\leq \mathsf{poly}(M) \cdot \left( \frac{L}{T} + \frac{1}{\sqrt{L}} + \rho + \frac{1}{\rho\sqrt{T}} \right)$$

where the four terms are:

(1) The difference between the optimal objectives on games $\Gamma$ and $\tilde{\Gamma}$. It is at most $2n\varepsilon$ because
$F^*(\Gamma) = F(\mu^*, P^*) \leq F(\mu^*, P^* + 2\varepsilon) + 2n\varepsilon \leq F(\tilde{\mu}^*, \tilde{P}^*) + 2n\varepsilon = F^*(\tilde{\Gamma}) + 2n\varepsilon.$

(2) The suboptimality and payments in the utility learning stage,

(3) The bonus payments to ensure strict incentive compatibility in the steering stage, and

(4) The suboptimality and payments in rounds on which $\boldsymbol{a}^t \neq \boldsymbol{s}^t$.

Setting $\rho = T^{-1/4}$ and $L = T^{2/3}$ then completes the proof.

