# OpenReview forum: "Learning a Game by Paying the Agents"
_ICLR.cc/2026/Conference — ICLR 2026 Poster_

### Official Review · Reviewer_2pKK · 2025-10-24

**Soundness:** 3
**Presentation:** 3
**Contribution:** 3
**Rating:** 6
**Confidence:** 3

**Summary:**

This paper studies an active, non-equilibrium inverse game problem interacts with any no-regret players by giving payments and signals. The obejctive is to learn the utility functions of players under a equivalent game. The main idea is to cast the principal–agent interaction as a zero‑sum game where the principal’s mixed strategy is a payment vector. Then, authors consdier the problem of steering no-regret learning algorithm to desired result without the access to utility functions of players.

**Strengths:**

- The zero‑sum formulation simple yet interesting, which requires almost no modeling of the agent’s learning rule beyond no‑regret property.

- The results of this paper hold for arbitrary no‑regret learner, and authors also provide a lower bound implying the tightness of $\epsilon$ and necessarity of $M$.

- The proposed approaches are extended to the problem of steering no-regret learner without the access of utility function. Authors show that the principal-optimal CEP characterizes the value that the principal can achieve.

- The paper is well-written and easy to follow. I appreciate that authors offer clear intuitions for the most of places.

**Weaknesses:**

- The assumption in (1) needs to hold for all signals, which seems strong to me. In fact, many contextual algorithms ensure provable regret bounds on average over contexts, not uniformly over all contexts.

- The sample complexity shown in Theorem 4.3 has polynomial dependence on M and C, but it does not match the given lower bound.

- The rate of steering agents to achieve the optimal CEP is $T^{-1/4}$, which seems suboptimal.

**Questions:**

- I would like to hear whether it is possible to relax the requirement on the signal in Theorem 4.3. Could the authors clarify or discuss potential directions for such relaxation?

- In Section 3, it is mentioned that the knowledge of the mixed strategy $x_t$ could be relaxed. Could the authors elaborate on how the proposed algorithm should be modified in that case?

---

> ### Author Response · Authors · 2025-11-21
>
> **(Weakness 1 & Question 1)** We don't think the assumption in (1) of no-regret for all signals is too strong.  Note that, unlike many contextual no-regret algorithms with an infinite, continuous context space, we only have a finite signal/context space: the principal recommends actions or sends $\bot$ to the agents, which means that the signal space is just as large as the agent's action set plus 1, which is finite.  An easy way to construct contextual no-regret algorithms for a finite context space is to just run regular no-regret algorithms for different contexts separately.
>
> Also, our use of signals is largely motivated by our main application: the steering problem.  Previous work [Zhang et al, EC 2024, Steering no-regret learners to a desired equilibrium] showed that the use of signals is necessary in steering: without signals, there exist games where the principal cannot steer no-regret learning agents to achieve the optimal Nash equilibrium with sublinear (in time horizon) payment.  Regardless of the steering problem, we agree that whether the utility learning problem (with multiple agents) can be solved efficiently without using signals is a very interesting future direction.  Despite several attempts, we don't have concrete results for this direction yet.
>
>
> **(Question 2)** The knowledge of the mixed strategy $x_t$ can be relaxed as follows: Consider the single-agent algorithm (Algorithm 1).  Suppose the principal only observes the realized action $a^t$ taken by the agent at each round. Let $1_{a^t} = (0, ..., 0, 1, 0, ... 0)$ be the indicator vector whose $a^t$-th component is 1.  We think of $1_{a^t}$ as the "mixed strategy" played by the agent in round $t$.  Then, in Algorithm 1, the principal does gradient descent by $p^{t+1} = \Pi[p^t - \eta 1_{a^t}]$; namely, just replace $x^t$ by $1_{a^t}$.  Because the agent's regret on the sequence of mixed strategies is $C\sqrt{T}$, by Azuma's inequality, the agent's regret on the sequence of realized actions is at most $C\sqrt{T} + O(\sqrt{T})$ with high probability.  Our argument seamlessly applies to no-regret on the realized actions, so the principal can still learn the game in $O(\frac{m^3 + C^2 m^2}{\varepsilon^2})$ rounds with high probability.  The same applies to the multi-agent case, where we just need to change the $x_i^t$ to $1_{a_i^t}$ in Algorithm 2, and have the same round complexity with high probability.

---

### Official Review · Reviewer_evWz · 2025-11-03

**Soundness:** 4
**Presentation:** 2
**Contribution:** 4
**Rating:** 6
**Confidence:** 4

**Summary:**

This paper studies the problem of learning a game by observing the agents play the game using no-regret algorithms and paying them to influence their behavior. The actor in this game seeks to learn the utilities of the game, up to strategic equivalence, through picking payments and providing signals to the agents in each round of play, under the assumption that the agents play arbitrary, possibly correlated, no-regret algorithms. It is important to note that the agents are in fact assumed to have contextual no-regret, i.e. they have vanishing regret for each signal/ no-regret over strategies mapping signals to actions. The main result of the paper shows that it is possible to learn the game in number of rounds polynomial in the size of the payoff matrix and that there are matching lower bounds, up to polynomial factors.

The paper first shows a slick method to estimate the utilities of a single agent using payments, by defining a two-player zero-sum game with a unique equilibrium and then shows how to construct no-regret dynamics to generate a sequence of payments that approaches this equilibrium, which in turn reveals the utility function of the agent.  This method is then lifted to learn the utilities of all agents in sequence in a multi-agent game, by rotating through the actions of all other players and using signals to fix their behavior for each learning phase. Other results in the paper include using this result to steer agents to desirable correlated equilibria while starting with no initial information about the game.

**Strengths:**

The main strength of the paper is in posing a natural question about learning a game by observing no-regret behavior and making meaningful progress in answering it by extending existing technical machinery. In particular, the algorithm to find the utility of a single agent is an elegant construction to re-use no-regret dynamics to find the equilibria of a game despite not seeing full information feedback.

**Weaknesses:**

The main weakness of the paper is the strong assumption made about the agents having contextual no-regret over the signals employed by the actor learning the game. This gives the actor tremendous leverage over the players practically for "free" since it allows them to "forget" the history of play by switching signals. This is exploited in obtaining an almost direct reduction from multiple players to one player. While this might well be a necessary assumption, there is insufficient technical justification for it.

**Questions:**

Connected to the main weakness highlighted above -- how necessary is the assumption about no-regret for each signal? For instance, is it at all possible to learn the utilities of a multiplayer game with just a vanilla external regret assumtion?

---

> ### Author Response · Authors · 2025-11-21
>
> Our use of signals is largely motivated by the main application of our results: the steering problem.  Previous work [Zhang et al, EC 2024, Steering no-regret learners to a desired equilibrium] showed that the use of signals is necessary in steering: without signals, there exist games where the principal cannot steer no-regret learning agents to achieve the optimal Nash equilibrium with sublinear (in time horizon) payment.  Regardless of the steering problem, we agree that whether the utility learning problem (with multiple agents) can be solved efficiently without using signals is a very interesting future direction.  Despite several attempts, we don't have concrete results for this direction yet.

---

### Official Review · Reviewer_bHow · 2025-11-05

**Soundness:** 2
**Presentation:** 2
**Contribution:** 2
**Rating:** 6
**Confidence:** 4

**Summary:**

Dear authors,
I am finalizing my evaluation and will upload my full review within the next two days. Thank you for your patience.

Best regards,
—Reviewer

**Strengths:**

TBC

**Weaknesses:**

TBC

**Questions:**

TBC

---

> ### Comment · Reviewer_bHow · 2025-11-17
> **Review submission**
>
> I would like to apologize for the delay in submitting my review. Thank you for your patience.

---

> ### Author Response · Authors · 2025-11-21
>
> Thanks for your detailed comments!
>
> **1: Concept of Signals/Context in No-Regret Algorithms**
>
> (1a) Exactly as you commented and as we mentioned in Line 193, one example of contextual no-regret algorithms is to run a regular no-regret algorithm (such as FTRL, MWU) for each context independently. For example, the contextual MWU algorithm maintains separate weight vectors for different signals and updates the weight vector only for the observed signal in each round, as you pointed out.  It is also possible that the algorithm can share information across different contexts.  As long as the algorithm satisfies no-regret for every context, our conclusion holds.
>
> (1b) Context/signals naturally appear in **games with communication**, where a principal provides information to players before players choose their actions. This communication idea dates back to correlated equilibrium [Aumann, 1974], is related to information design in economics [Kamenica & Gentzkn, 2011, "Bayesian Persuasion"], and has recently been studied by the "Game Theory + Machine Learning" community as well (e.g., [Lin et al, NeurIPS 2023, "Information Design in Multi-Agent Reinforcement Learning"]).
>
> A concrete example is advertising auctions where bidders use learning algorithms to choose bids.   Because bidders are uncertain about the ever-changing environment, they use adversarial no-regret algorithms.  The seller knows the clickthrough rate of the ad slot for sale, and provides an estimation of the clickthrough rate as a signal to influence bidders' bidding decisions.  This model has been considered by, e.g., [Bergemann et al, WebConf 2022, "Calibrated click-through auctions"].  Such an example features both no-regret learners and signals.
>
>
> **2: Why Should the Agents Obey the Signals?**
>
> *We do not require the agents to obey the signals*.  The agents run no-regret algorithm to choose actions in response to each context/signal, treating the environment as adversarial as you pointed out. If a signal (i.e., recommended action) is far from optimal, then the agent will figure that out eventually and not take that action. But our Algorithm 2 guarantees that, whenever the principal recommends an action to the agent, it is indeed optimal (no-regret) for the agent to obey that recommendation.  We prove this fact formally in the proof of Theorem 4.3; intuitively, it is because the principal provides a large payment to the agent if the agent takes the recommended action.
>
> The agents are not collaborating with the principal.  The principal needs to provide payments to incentivize the adversarially no-regret learning agents to obey the signals.
>
>
> **3. Proposition B.1 and the Definition of "Any-Time"**
>
> The statement is the following: Suppose that, for some fixed values of $T$ and $B$, some no-regret algorithm guarantees that, against any adversary, its regret $R_i(T)$ after $T$ time steps is bounded by $B$. A priori, it could be conceivable that the algorithm might allow the regret to exceed $B$ at some point $t < T$, and then play in such a way that the regret then drops below $B$ by time $T$. The proposition states that this, in fact, cannot happen.
>
>
> **4. Line 221 - Why Do All No-Regret Algorithms Ignore Additive Constants**
>
> Indeed, some no-regret algorithms depend on the absolute values of the feedback.  However, the definition of regret only depends on the utility difference between two actions.  All our conclusions (both utility-learning and steering results) hold as long as the agent's algorithm satisfies no-regret, which includes algorithms depending on absolute values.  We have revised Line 221 to "agents' regrets are only affected by the utility differences between actions, and the behaviors of typical no-regret algorithms (such as MWU) only depend on those differences."
>
> **5. The “Obedience” Problem Revisited: Why Restart at a Signal?**
>
> As we mentioned in Question 2, agents do not simply obey the signals.
>
>
> **6. No-Regret Against an Adversary – What Is the Realism of This Setting?**
>
> Please see (1b) for a concrete example.

---

> > ### Comment · Reviewer_bHow · 2025-11-21
> > **2: Why Should the Agents Obey the Signals - Follow-up questions**
> >
> > I understand the authors’ point that in principle agents do not need to “obey” the signals, and that incentive compatibility is intended to come from the payment structure. However, even with payments, I am still confused about why a no-regret algorithm would decide to follow different signals and effectively run different no-regret instances across contexts.
> >
> > A key issue is the following:
> > Suppose I am an agent running a single adversarial no-regret algorithm that is oblivious to context. Then my regret guarantee holds regardless of whether the principal wants to separate the interaction into different contextual sub-periods. In fact, from my perspective as a learner, refusing to switch contexts and simply continuing with my standard algorithm still yields vanishing regret. This seems strictly worse for the principal and strictly neutral (or better) for me.
> >
> > Let me restate the concern more sharply:
> > 	•	A no-regret learner does not benefit from resetting or switching contexts.
> > 	•	Algorithms like FTRL are “forgivingness-free”: they accumulate history and are intentionally resistant to manipulation.
> > 	•	Therefore, the principal appears to face a structural disadvantage—if they want to reinitialize learning dynamics, they must pay for long enough to drag the algorithm back toward something close to a uniform prior before a restart becomes effective.
> >
> > In other words, even with payments, it is unclear why behaviorally the learner would ever choose to run separate contextual no-regret processes when sticking to a single, oblivious no-regret algorithm is sufficient and adversarially optimal.
> >
> > Could the authors clarify how their model overcomes this apparent tension?

---

> ### Author Response · Authors · 2025-11-21
>
> **7. Lower Bound Argument – The C / (2\sqrt{T}) Issue**
>
> Our assumption is that the agent's algorithm's regret is at most $C\sqrt{T}$, the principal knows $C$ but no additional details about the algorithm. The $C$ is not necessarily the optimal constant, because agents in practice might run sub-optimal algorithms, such as MWU with a sub-optimal learning rate, for various reasons.  Also, the agent might be running a better algorithm than the principal expected, with regret smaller than $C\sqrt{T}$, like $C/2\sqrt{T}$, but the principal is unaware of that.
>
> We clarify our indistinguishability argument in the lower bound proof (Appendix C.2).  Consider an agent running an algorithm with at most $C\sqrt{T}$ regret on utility function $U_i$.  Consider another agent running a better algorithm with regret at most $C/2 \sqrt{T}$ but on utility function $U_i + Z_i$ where $Z_i$ and $U_i$ differ by $\varepsilon = \frac{C}{2\sqrt{T}}$.  Then, the second agent's regret with respect to $U_i$ is also at most $C\sqrt{T}$, same as the first agent.  We can construct the utility functions and the no-regret algorithms in a way that the two agents' behaviors are exactly the same.  Hence, the principal cannot distinguish between these two utility functions $U_i+Z_i$ and $U_i$, meaning that the principal cannot learn the utility function with accuracy better than $\epsilon < \frac{C}{2\sqrt{T}}$.
>
>
> **8. Algorithm 2 - Why is Line 7 inside the third loop?**
>
> Note that the third loop is the enumeration over the $L$ rounds in a phase, while the first loop over agents and the second loop over action profiles enumerate phases but not rounds.  In our model, the principal sends a signal to every agent at every round, and importantly, different signals can be sent at different rounds.  So the signal-sending step should be inside each round, which is inside the third loop.  It is true that our algorithm actually sends the same signal to an agent during the $L$ rounds in a phase, so we can move Line 7 to between the second and third loops (Line 4), but then we need to write "principal sends signals $s_i^t = \bot$ and $s_j^t = \bar a_j$ for every $j \ne i$ **for the next $L$ rounds**", which might be confusing.
>
>
> **9. Proof of Proposition D.5 and D.2.1**
>
> Note that Proposition D.5 is about "signal-dependent payments" vs "signal-independent payment", instad of "non-entangled signals" vs "entangled signals".  We have provided the missing details for the proof of Proposition D.5 in Appendix D.2.1.  Please find the deails in the revised PDF and let us know if more clarifications are needed.
>
>
> **10. Steering and the Role of No-Swap Regret**
>
> (10a & 10b) In the steering section, our positive result (Theorem 5.3), which shows that the principal can steer no-regret learners to the optimal CEP, holds for all no-regret learners, including no-swap-regret learners.  Our paper indeed focuses on general no-regret learners, for which this positive result holds.
>
> Our negative result (Theorem 5.2) says that the principal cannot do better than the optimal CEP for some no-swap-regret learners.  Since no-swap-regret learners are also no-regret, this theorem directly implies that there exist no-regret learners such that the principal cannot do better than the optimal CEP.  Restricting to no-swap-regret learners in fact makes our negative result stronger.
>
> And as we mentioned in Q2, our agents do not necessarily obey the signals.  They run no-regret algorithms to learn to respond to signals.  All of our results hold under that assumption.
>
> (10c) While the convergence of no-swap-regret learning to correlated equilibrium is a standard result, there are two subtleties.  (1) The standard result is for normal-form games with no payments, where the signals sent by a correlating device do not affect players' payoffs -- the signals only affect players' beliefs about other players' strategies.  In our games with payments, the payment depends on the players' actions as well as the signals, so our signals directly affect the players' payoffs. Our CEP notion is thus different from the classical correlated equilibrium notion, and the convergence of no-swap-regret learning to CEP requires a new proof.  (2) A game may have multiple CEPs. No-swap-regret learning converges to one of them, not necessarily the optimal one.  Our result (Theorem 5.3) shows that the principal can steer no-swap-regret learners to converge to the optimal CEP.
>
>
> (10d) Just like contextual no-regret algorithms, a typical way to construct contextual no-swap-regret algorithms is to instantiate a regular no-swap-regret algorithm (such as [Blum & Mansour, JMLR 2007, From External to Internal Regret]) for each context, and run those algorithms independently.  This type of contextual no-swap-regret algorithms has been studied by previous work on learning in games with communication (e.g., [Lin & Chen, ICLR 2025, Generalized Principal-Agent Problem with a Learning Agent]).

---

> > ### Comment · Reviewer_bHow · 2025-11-21
> > **7. Lower Bound Argument – The C / (2\sqrt{T}) Issue**
> >
> > I would like to here practical reasons that a player is not choosing the optimal constants.

---

> > ### Comment · Reviewer_bHow · 2025-11-21
> > **9. Proof of Proposition D.5 and D.2.1**
> >
> > Please believe that part of the delay in my response comes from my genuinely trying to work through the calculations of appendix myself. Personally, I would very much appreciate it if the authors could provide fully detailed analytic derivations for both cases, in a follow-up note, so that one can reproduce all intermediate steps. Having these calculations written out extensively would make me feel much more confident about the technical claims.

---

> > > ### Author Response · Authors · 2025-11-24
> > >
> > > > Let me restate the concern more sharply: • A no-regret learner does not benefit from resetting or switching contexts. • Algorithms like FTRL are “forgivingness-free”: they accumulate history and are intentionally resistant to manipulation. • Therefore, the principal appears to face a structural disadvantage—if they want to reinitialize learning dynamics, they must pay for long enough to drag the algorithm back toward something close to a uniform prior before a restart becomes effective.
> > >
> > > This is, in essence, the reason that our analysis requires signals.
> > >
> > > > In other words, even with payments, it is unclear why behaviorally the learner would ever choose to run separate contextual no-regret processes when sticking to a single, oblivious no-regret algorithm is sufficient and adversarially optimal.
> > >
> > > Another possible response is the following: if the learner *is* receiving signals (which our learners are), why *shouldn't* the learner use this information in selecting its actions? After all, the learner can get a higher utility by using the signals than by not using the signals. And, running no-contextual-regret processes is the most natural way of formalizing what it means for a no-regret learner to make use of signals.
> > >
> > > As we have stated, we believe that the case where signals are disallowed is an interesting future direction.
> > >
> > > > I would like to here practical reasons that a player is not choosing the optimal constants.
> > >
> > > Depending on what the reviewer is intending with the question, we hope that one of the following two perspectives answers it:
> > >
> > > One way to think about the lower bound is the following. Our upper bound (Theorem 4.3) applies for any no-regret algorithm, and has dependence polynomial in $\text{poly}(M, C)/\epsilon^2$. A natural question to ask is whether it is possible to strengthen any of the dependencies. That is, can the $\text{poly}(M, C)/\epsilon^2$ in the theorem statement be replaced by some other function $T(n, m, M, C, \epsilon)$ that is much better, and still remain true? Theorem 4.3 provides a negative answer to this question: any such function $T$ must satisfy $T(n, m, M, C, \epsilon) \ge \Omega(n M) \log(1/\epsilon)$ and $T(n, m, M, C, \epsilon) \ge C^2/4\epsilon^2$. This framing of the lower bound question completely removes any concern about how the agent picks an algorithm.
> > >
> > > One other possible question that the reviewer might be asking is this: if $C$ is the best possible constant achievable by an adversarial no-regret learning algorithm, how is the result still true? There is no adversarial no-regret algorithm with regret guarantee $C\sqrt{T}/2$ by assumption, so one might think that the proof breaks here. This, however, is not the case, and the reason gets at a subtle corollary of our modeling choices: we never assume in the paper that the learners are actually running any "natural" no-regret learning dynamics per se, only that they *end up*, in hindsight, having the anytime no-regret property. Therefore, for example, one valid instantiation of the "learning algorithms" required by the proof of Theorem 4.4 is one in which all agents, in every round, play a Nash equilibrium of the game with payments and noises, i.e., the game with utility functions $U_i + Z_i + P^t_i$. Such agents incur only $\epsilon T$ regret asymptotically.
> > >
> > > As we stated in the paper and also in response to another reviewer, our main motivation for studying this problem is to apply it to the problem of steering no-regret learners [Zhang et al. 2024]. Their paper also requires and uses signals in a similar manner to our paper, and also uses a similar "no regret in hindsight" assumption that allows the agents to do things other than run adversarial no-regret algorithms. So, none of these modeling choices are unique to our paper.
> > >
> > > If these answers are satisfactory, we are happy to make a revision that explicitly mentions these points.
> > >
> > > > Please believe that part of the delay in my response comes from my genuinely trying to work through the calculations of appendix myself. Personally, I would very much appreciate it if the authors could provide fully detailed analytic derivations for both cases, in a follow-up note, so that one can reproduce all intermediate steps. Having these calculations written out extensively would make me feel much more confident about the technical claims.
> > >
> > > Such a derivation for Prop. D.5 is already included in the revised version of the paper, marked in blue and much more detailed than before. We welcome any further questions.

---

### Official Review · Reviewer_zfeR · 2025-11-12

**Soundness:** 4
**Presentation:** 3
**Contribution:** 3
**Rating:** 6
**Confidence:** 3

**Summary:**

This game addresses a natural question of learning a player's utility functions over time and studies this from an online learning perspective. Although this is a standard problem, the paper differs from the literature in two important ways: first they assume that the players are not in equilibrium. Secondly, they assume that the players are playing a no regret algorithm. The main contribution is a polynomial time algorithm that learns the utilities upto an error of $\epsilon$. They further show that they can guide the agent towards a particular correlated equilibrium with payments (also known as steering).

The model is as follows. In each round the principal payment functions $P^t_i: A_i \to [0,B]$  gets added to the agent rewards. The principal can also send a private signal. The agent which maximizes the sum of the payment and their own utility will play a mixed strategy $x^t$ which is observed by the principal. The paper's central insight is to reduce this to a zero-sum game between the principal and agent. Effectively the unique optimal in this game is for the principal to offer a payment of $p = 1-u$ the utility, thus making the agent indifferent between the actions. Then the principal can read the utility function by simply taking $p=1-u$. This is because the utility is zero-mean and if the principal updates their weights according to a no-regret algorithm like projected gradient descent (note that the mixed strategy $x^t$ is the derivative of the principal's loss function $(u+p) \cdot x^t$), it will eventually converge to a point where the optimal utility is $u+p$ is a constant. This is a simple but elegant idea that allows the principal to discover the agent's true utility.

To handle the general setting when there are $n$ agents, the principal uses signals to effectively freeze the other agent's actions.

**Strengths:**

I think the paper outlines a simple and beautiful idea to learn the utility of any agent in a normal formed game. The dependence and the  learning rate and the associated regret is tight.

**Weaknesses:**

The complexity to learn scales exponentially on the number of agents and their action profiles. Their lower bound suggests that this task is too hard for many practical games. I also wonder a conference such as EC  might be a better fit than ICLR.

**Questions:**

N/A

---

> ### Author Response · Authors · 2025-11-21
>
> Thanks for your comments!  We appreciate your assessment that our paper "outlines a simple and beautiful idea" for the utility learning problem.  We believe a paper on inverse game theory and no-regret learning fits both EC and ICRL.  Our approach of using payments to learn a game might be interesting to e.g., the inverse reinforcement learning community on ICRL.

---

### Meta-Review · Area_Chair_UUHZ · 2026-01-07

**Summary:**

The paper studies an active inverse game theory problem in repeated normal-form games. In particular, the authors consider a setting where principal observes play by no-regret agents and is additionally allowed to (i) send private signals and (ii) add nonnegative action-dependent payments to agents' rewards.

The key technical contribution is a reduction of single-agent utility learning to finding an equilibrium in a two-player zero-sum game where the principal's mixed strategy is a payment vector. By running no-regret dynamics on both sides, the average payment converges to the negative utility, revealing the agent's utility up to strategic equivalence. The paper then lifts this idea to multi-agent games using signals to "separate" phases and effectively hold other agents' behavior fixed while learning one agent's conditional utilities, and finally uses the learned game as a subroutine to steer learners toward a principal-optimal correlated equilibrium with payments.

All reviewers find the single-agent core idea technically clean and conceptually appealing, and the results are generally assessed as sound.

At the same time, there is a shared concern that the multi-agent learning/ steering results lean heavily on a strong contextual no-regret assumption over signals (effectively giving the principal significant leverage by switching contexts), and there are also practicality/ scale concerns since the round complexity is polynomial in the normal-form size, which is exponential in the number of agents for constant action sets.

Overall, after taking the rebuttal and discussion into account, I view the paper as a solid contribution with an interesting proof technique (zero-sum formulation and no-regret convergence), but with an important modeling caveat around contextual regret that should be emphasized and better motivated.

**Reviewer Concerns:**

Addressed

* Clarification of contextual no-regret and obedience
* Motivation for signals via steering
* The authors explain how to adapt the algorithm when only realized actions are observed
* suboptimal constants in lowerbound -- the authors emphasize that their modeling only requires an "in-hindsight" anytime no-regret property rather than a specific canonical algorithm with optimal tuning.


Still outstanding
* The assumption of no-regret within each context can effectively allow the principal to reset learning dynamics by switching signals. The paper does not provide a weaker alternative assumption under which multi-agent learning remains efficient
* the final paper should still make the modeling assumptions and their implications more explicit up front.

**Reviewer Scores:**

Most likely reviewers will maintain their scores.

---

### Decision · Program_Chairs · 2026-01-26

Accept (Poster)